# Food insecurity is associated with decreased quality of life in patients with chronic Chagas disease

Celson Júnio do Nascimento Costa[1], Paula Simplício da Silva[2],
Roberto Magalhães Saraiva[2], Luiz Henrique Conde Sangenis[2],
Marcelo Teixeira de Holanda[2], Gilberto Marcelo Sperandio da Silva[2],
Fernanda de Souza Nogueira Sardinha Mendes[2], Isis Gabrielli Gomes Xavier[2],
Henrique Silveira Costa[3], Tatiana Rehder Gonçalves[4], Luiz Fernando Rodrigues Junior[1],
Cristiane da Cruz Lamas[1,2], Grazielle Vilas Boas Huguenin[1],
Alejandro Marcel Hasslocher-Moreno[2], Daniel Arthur Barata Kasal[1],
Mauro Felippe Felix Mediano[1,2]*

1 Department of Research and Education, National Institute of Cardiology, Rio de Janeiro, RJ, Brazil,
2 Evandro Chagas National Institute of Infectious Disease, Oswaldo Cruz Foundation, Rio de Janeiro, RJ,
Brazil, 3 Department of Physical Therapy, Federal University of Vales do Jequitinhonha e Mucuri, Minas Gerais,
MG, Brazil, 4 Institute of Collective Health, Federal University of Rio de Janeiro, Rio de Janeiro, RJ, Brazil

* mffmediano@gmail.com, mauro.mediano@ini.fiocruz.br

## Abstract

This study assessed the prevalence of food insecurity (FI) in patients with Chagas Disease (CD) and its association with anthropometric measurements, comorbidities, and quality of life (QoL). This cross-sectional study included CD patients of both sexes. The FI was evaluated using the Brazilian FI scale. Anthropometric measurements included weight, height, and waist circumference. Comorbidities included hypertension, diabetes, dyslipidemia, and obesity. Lipids and plasma glucose were collected. QoL was assessed using the Portuguese version of WHOQOL-Bref questionnaire. Linear and logistic regression models were fitted to evaluate the association between FI status and outcomes. From the 359 included participants (55.9% women, median age 62 years), 22.5% had mild, 5.6% moderate, and 2.5% had severe FI. In the adjusted models, mild FI was significantly associated with an increased odds of obesity (OR=1.83, 95%CI=1.03 to 3.25). Moreover, significant associations were observed between FI and all QoL domains, including physical health (low FI: β=−8.43 95%CI −12.36 to −4.49; moderate/severe FI: β=−8.02 95%CI −14.11 to −1.94), psychological (low FI β=−5.54 95%CI −9.11 to −1.96; moderate/severe FI β=−7.22 95%CI −12.76 to −1.68), social relationship (low FI: β=−6.62 95%CI −10.37 to −2.88), environment (low FI: β=−8.79 95%CI −11.99 to −5.60; moderate/severe FI β=−13.56 95%CI −18.51 to −8.62), and overall (low FI β=−8.10 95%CI −12.15 to −4.06; moderate/severe FI: β=−16.82 95%CI −23.08 to −10.57). FI was consistently associated with poor QoL in patients with chronic CD.

**Data availability statement:** The dataset related to the present study is available in the Open Science Framework database (https://osf.io/9ev8b/).

**Funding:** The author(s) received no specific funding for this work.

**Competing interests:** The authors declare that they have no competing interests.

## Introduction

Chagas disease (CD) is an infectious disease caused by the parasite *Trypanosoma cruzi* that affects about 6–8 million individuals worldwide, with most cases concentrated in Latin America [1]. Migratory movements have contributed to the increase in the number of CD cases in regions beyond Latin America, including North America and Europe [2]. Despite progress in reducing vector transmission, particularly in rural areas, CD remains a persistent and neglected disease, with many affected individuals facing barriers to timely diagnosis and treatment [3]. Additionally, the migration of a significant proportion of the CD population from rural to urban areas has led to greater exposure to unhealthy lifestyles, which may contribute to the development of several non-communicable diseases, such as obesity, hypertension, diabetes mellitus, and dyslipidemia, while negatively affecting their quality of life (QoL) [4,5].

CD is usually associated with poverty, low socioeconomic status, and low educational levels [6,7], being considered as one of the neglected tropical diseases that disproportionately affect people living in poverty by the World Health Organization (WHO). CD presents significant challenges for both patients and healthcare systems, particularly among socially vulnerable communities, exacerbating physical and psychological distress while further amplifying socioeconomic inequities. The factors that contribute to the link between CD and poverty are complex, including poor housing conditions and inadequate access to healthcare services, that are more commonly observed in middle and low-income countries [8]. Conversely, in more severe cases, CD can result in work incapacity, increasing poverty. Moreover, the treatment costs of the cardiac and digestive clinical forms of CD and the distance to healthcare services may increase the financial burden of the affected individuals and their families [9–11].

Considering their low socioeconomic status, individuals with CD are more vulnerable to food insecurity (FI) — a condition characterized by limited access to sufficient, safe, and nutritious food needed to meet dietary requirements and maintain a healthy, active lifestyle [12]. Previous studies have demonstrated that FI is negatively associated with several health outcomes, such as increased risk of obesity and other chronic diseases [13,14] and mental health problems [15]. In addition, different studies have shown that FI may be associated with a decreased QoL [16–18]. Given the potential intersection of FI and the heightened vulnerability of individuals with CD, the investigation of FI and its association with comorbidities and QoL in individuals with CD is of paramount importance [19]. To the best of our knowledge, no previous study has investigated FI in CD population. Therefore, the present study aimed to assess the frequency of FI in an urban cohort of patients with CD and its association with anthropometric measurements, lipid profile and blood glucose, comorbidities, and QoL.

## Materials and methods

### Ethics approval and consent to participate

The present study was approved by the Ethics Committee in December 2013, in accordance with the resolution 466/2012 of the Brazilian National Council of Health

(CAAE: 22985313.8.0000.5262). All participants received information about the goals and procedures of the study and agreed to participate by signing an informed consent form.

## Study design and population

This is an observational cross-sectional study, conducted from March 2014 to March 2017. The study population consisted of patients diagnosed with CD, confirmed by positive serology through two different methods (ELISA and indirect immunofluorescence), regularly followed at the Evandro Chagas National Institute of Infectious Diseases (INI), a unit of the Oswaldo Cruz Foundation (Fiocruz) located in Rio de Janeiro, Brazil. The INI/Fiocruz is a national reference center for the treatment and research of infectious and tropical diseases within the Brazilian National Health Service (Sistema Único de Saúde – SUS), operating under the Brazilian Ministry of Health. The INI/Fiocruz receives patients from various regions across the country, providing comprehensive, multidisciplinary care to patients with CD [20].

Patients with confirmed diagnosis of CD from both sexes and aging ≥ 18 years were consecutively enrolled during their routine outpatient visits at INI-Fiocruz. Patients with non-chagasic heart disease, immune system disorders, cancer, other infectious diseases during the data collection, those under use of corticosteroids or anti-inflammatory drugs within the previous three months of data collection, those with severe cognitive impairments, and pregnant women were excluded.

## Study procedures

Patients were invited to participate in the study during their regular medical appointments. The study procedures were carried out in two visits within a period of no more than two months between visits. During the first visit, patients signed the informed consent, completed all the questionnaires, including QoL and FI, and underwent anthropometric measurements. In the second visit, patients underwent a clinical evaluation (including electrocardiogram and echocardiogram) and collected blood tests. Trained staff administered the questionnaires and performed the anthropometric measurements. The same physician performed the clinical evaluation in all participants.

## Food insecurity

FI was assessed using the Brazilian Scale of Food Insecurity (EBIA), which had been previously validated and was available at the time of the study's data collection [21]. The EBIA questionnaire consisted of 15 questions with yes or no responses, aimed at assessing the experience of food insufficiency over the last three months. For each affirmative answer, one point was assigned, and the total score comprised the sum of the overall questions, ranging from 0 to 15 points. The interpretation of the score was based on the household composition. For households with residents <18 years old, no FI corresponded to a score of 0, mild FI corresponded to a score of 1–5 points, moderate FI corresponded to a score of 6–10 points, and severe FI corresponded to a score of ≥11 points. For households with only adults (≥ 18 years old), no FI corresponded to a score of 0, mild FI corresponded to a score of 1–3 points, moderate FI corresponded to a score of 4–6 points, and severe FI corresponded to a score of ≥ 7 points.

## Anthropometric measurements

Body weight was measured using a Filizola® digital scale (Filizola, São Paulo, Brazil), with a variation of 0.1 kg and maximum capacity of 150 kg. Height was measured by double measurement using a stadiometer accoplated to Filizola® digital anthropometric scale, with a maximum allowable variation of 0.5 cm between measurements. Body mass index (BMI) was calculated using the following formula: BMI = body weight (in kg) divided by height (in meters) squared. Waist circumference was measured at the point of minimum circumference between the last rib and the iliac crest [22]. Double measurements were taken, with a maximum allowable difference of 1 cm, and the average value of the two measurements was used.

## Comorbidities

History of arterial hypertension, diabetes, and dyslipidemia were assessed by the same physician by medical interview, blood tests, and medical records review. Obesity was defined based on anthropometric measures if BMI (weight/height$^2$) was ≥ 30 kg/m$^2$ [23].

## Lipid profile and glucose

Blood samples were drawn in the morning after a 12-hour fasting period. Plasma lipids were measured using a commercial kit (Gold Analisa®, Belo Horizonte/MG, Brazil). LDL and VLDL cholesterol were estimated using the Friedewald formula [24]. Plasma glucose was determined through enzymatic-colorimetric assay (GoldAnalisa kit), using a Konelab 6.0.1 device, with automated reading at a wavelength of 500nm.

## Quality of life

The assessment of quality of life (QoL) was conducted using the WHOQOL-Bref questionnaire validated for the Brazilian population [25,26]. The WHOQOL-Bref is a generic questionnaire comprising 26 items, two of those relating to the overall perception of QoL (referred to as the overall domain) and the remaining 24 items covering four distinct domains of QoL (physical, psychological, social relationships, and environment). Each domain was evaluated by assigning question scores ranging from 1 (none) to 5 (very much) and the mean score of each domain was calculated. A higher mean score indicates a more positive QoL perception.

## Covariates

Sociodemographic, clinical, and lifestyle covariates were obtained to characterize the study population and to address the potential confounding for the associations between FI with comorbidities and QoL. Information on age, sex, schooling, region of origin, race, and income per capita were collected during the interviews [27]. Age was considered as a continuous variable and was calculated by subtracting the date of birth from the date of the interview. Schooling was classified based on the formal years of study as <9 years, 9–12 years, and >12 years. The region of origin was determined by asking participants a single question about the state of their birth and was then categorized into either the northeast or other regions. Race was self-reported and re-classified as white and non-white (black, mulatto, yellow and Indigenous). Income per capita was determined by dividing the total income of all residents in the domicile by the number of residents. The classification of the clinical form of CD was obtained by the same physician after reviewing medical records and analyzing the electrocardiogram and the echocardiogram performed on the second visit and followed the recommendations made by the Brazilian Consensus on Chagas Disease [28]. To facilitate data analysis and following a clinical reasoning, patients were recategorized into indeterminate, cardiac without heart failure, cardiac with cardiac failure and digestive form. Food consumption was assessed using a 24-hour recall that consists on the identification and quantification of all food and beverages consumed in the day before the interview [29]. Macronutrients were calculated using DietWin Professional Version 2008 software.

## Data analysis

Exploratory data analysis was conducted by calculating the median (25%−75% interquartile range) for continuous and percentage (frequency) for categorical variables. Individuals with incomplete data were excluded from the analysis. Moderate and severe FI were grouped to facilitate data analysis due to the small number of participants within these groups. Differences across categories of FI were tested using Kruskal Wallis test for continuous and Chi-squared test for categorical variables.

To assess the association between FI categories (exposure) and the presence of comorbidities (arterial hypertension, diabetes mellitus, dyslipidemia, and obesity) as the outcome, we used logistic regression models to estimate odds ratios (OR) with 95% confidence intervals (CI). For continuous outcomes, including anthropometric measurements, lipid profile, blood glucose, and QoL, we used linear regression models to estimate beta coefficients (β) with their respective 95% CIs. Models were fitted without adjustments and adjusted for age, sex, race, education, and the presence of cardiac and digestive forms of CD; however, for QoL, the adjusted model further included comorbidities (arterial hypertension, diabetes mellitus, dyslipidemia, and obesity) given their potential role as a confounder for the association between FI and QoL [30].

Data management was carried out using the Research Electronic Data Capture (REDCap) web application. Statistical analyses were performed using Stata 17.0 software, with statistical significance level set at 5%.

## Results

Of the 397 patients included in the study, 38 were excluded according to the pre-specified exclusion criteria, with the remaining 359 patients being included in the analysis (Fig 1). The overall characteristcs of the participants of the study are described in Supplemental Table 1. Table 1 depicts the major characteristics of participants included in the study stratified by FI status. The frequency of FI was 30.6%, with 22.6% reporting mild FI, 5.6% moderate, and 2.5% had severe FI (8.1% moderate/severe FI). Overall, those with mild and moderate/severe FI were more likely to be women, with lower educational levels, lower per capita income, eutrophic, and had lower QoL scores for all domains. Conversely, those with mild FI presented a higher caloric consumption and a higher carbohydrate intake.

The association between FI with health outcomes and QoL are demonstrated in Table 2. In adjusted models, mild FI was significantly associated with an increased odds of obesity (OR=1.83, 95%CI=1.03 to 3.25). In addition, there were significant associations between mild and moderate/severe FI with decreased QoL. Mild FI was negatively associated with physical health domain (β=−8.43, 95%CI=−12.36 to −4.49), psychological domain (β=−5.54, 95%CI=−9.11 to −1.96), social relationship domain (β=−6.62, 95%CI=−10.37 to −2.88), environment domain (β=−8.79, 95% CI=−11.99 to −5.60), and overall QoL domain (β=−8.10, 95% CI=−12.15 to −4.06), whereas moderate/severe FI was negatively associated with physical health domain (β=−8.02, 95% CI=−14.11 to −1.94), psychological domain (β=−7.22, 95% CI=−12.76 to −1.68), environment domain (β=−13.56, 95% CI=−18.51 to-8.62), and overall QoL domain (β=−16.82, 95% CI=−23.08 to −10.57).

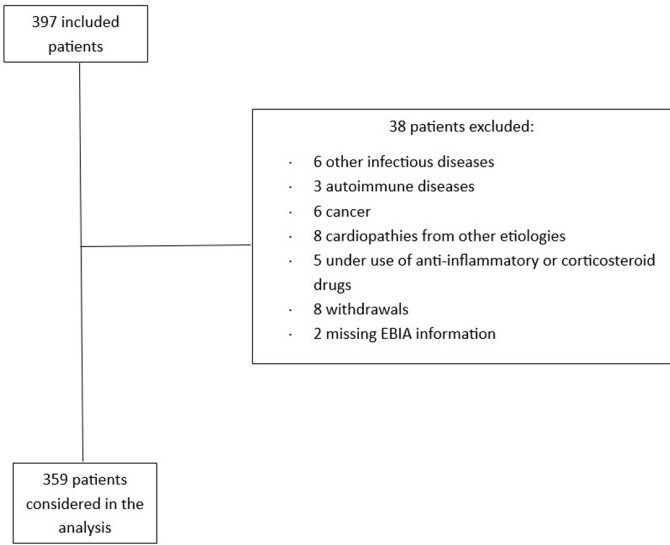

**Fig 1. Study flowchart.**

**Table 1. Clinical and demographic characteristics of participants stratified by FI status (n = 359).**

| Variables | No FI | Mild FI | Moderate/Severe FI | p-value* |
|---|---|---|---|---|
| | (69.4%; n = 249) | (22.5%; n = 81) | (8.1%; n = 29) | |
| **Age (Years) (Median; IQR 25% − 75%)** | 63 (54–69) | 60 (53-65) | 62 (58–65) | 0.15 |
| **Sex (%; n)** | | | | |
| Women | 52.6% (131) | 59.2% (48) | 75.9% (22) | 0.04 |
| **Race (%; n)†** | | | | |
| Non-white (*vs* white) | 78.3% (195) | 71.6% (58) | 86.2% (25) | 0.23 |
| **Schooling (%; n)** | | | | |
| <9 years | 64.2% (160) | 70.3% (57) | 89.7% (26) | 0.05 |
| 9–12 years | 19.7% (49) | 19.8% (16) | 3.4% (1) | |
| >12 years | 16.1% (40) | 9.9% (8) | 6.9% (2) | |
| **Region of origin (%; n)** | | | | |
| North | 1.2% (3) | 0% (0) | 0% (0) | 0.96 |
| Northeast | 67.1% (167) | 71.6% (58) | 72.4% (21) | |
| Southeast | 26.9% (67) | 25.9% (21) | 24.1% (7) | |
| South | 2.4% (6) | 1.2% (1) | 3.4% (1) | |
| Central west | 1.6% (4) | 1.2% (1) | 0% (0) | |
| Other countries‡ | 0.8% (2) | 0% (0) | 0% (0) | |
| **Nutritional status (%; n)** | | | | |
| Underweight | 1.2% (3) | 6.2% (5) | 0.0% (0) | 0.004 |
| Eutrophic | 30.9% (77) | 23.5% (19) | 48.3% (14) | |
| Overweight | 45.5% (113) | 35.8% (29) | 24.1% (7) | |
| Obesity | 22.5% (56) | 34.6% (28) | 27.6% (8) | |
| **Macronutrients intake (Grams) (Median; IQR 25% − 75%)** | | | | |
| Carbohydrate | 177.7 (143–241.4) | 186.6 (130.5–247.6) | 151.6 (106.8–200.5) | 0.05 |
| Lipid | 35.0 (26.6–50.3) | 34.7 (23.5–47.9) | 29.4 (17.4–44.4) | 0.22 |
| Protein | 63.0 (48.5–84.6) | 62.7 (44.6–73.5) | 57.5 (33.1–77.5) | 0.18 |
| Fiber | 16.7 (11.3–23.2) | 16.0 (12–24.6) | 13.7 (6–23.1) | 0.33 |
| **Caloric consumption (Kcal) (Median; IQR 25% − 75%)** | 1211.5 (895.6–612.9) | 1266.1 (977.6–1751.9) | 877.5 (689–1278.7) | 0.03 |
| **Hypertension (%; n)** | 67.0% (167) | 67.9% (55) | 65.5% (19) | 0.97 |
| **Diabetes Mellitus (%; n)** | 22.1% (55) | 22.2% (18) | 17.2% (5) | 0.82 |
| **Dyslipidemia (%; n)** | 52.2% (130) | 56.8% (46) | 58.6% (17) | 0.66 |
| **Chagas disease forms**** | | | | |
| Indeterminate form (%; n) | 29.3% (73) | 25.9% (21) | 10.3% (3) | 0.09 |
| Cardiac form without heart failure (%; n) | 53.4% (133) | 55.6% (45) | 58.6% (17) | 0.84 |
| Cardiac form with heart failure (%; n) | 14.1% (35) | 14.8% (12) | 27.6% (8) | 0.15 |
| Digestive form (%; n) | 14.9% (37) | 18.5% (15) | 17.2% (5) | 0.72 |
| **Per capita family income (R$) (Median; IQR 25% − 75%)** | 800 (550–1200) | 600 (400–800) | 375 (285–450) | <0.001 |
| **BMI (Kg/m²) (Median; IQR 25% − 75%)** | 26.5 (24–29.7) | 26.9 (24.2–31.8) | 26.9 (22.5–31.2) | 0.50 |
| **Weight (Kg) (Median; IQR 25% − 75%)** | 68 (58.8–79.5) | 69 (61.2–79.4) | 62 (58.8–72.5) | 0.23 |
| **Height (m) (Median; IQR 25% − 75%)** | 1.6 (1.54–1.66) | 1.57 (1.53–1.63) | 1.56 (1.51–1.62) | 0.06 |
| **Waist circumference (cm) (Median; IQR 25% − 75%)** | 90 (82.5–97.9) | 91.25 (80–100) | 86.5 (79.5–95.1) | 0.36 |
| **Total cholesterol (mg/dl) (Median; IQR 25% − 75%) n = 353** | 185 (160–205) n = 243 | 185 (165–210) n = 81 | 177 (158–194) n = 29 | 0.54 |
| **Triglycerides (mg/dl) (Median; IQR 25% − 75%) n = 352** | 103 (67–142) n = 242 | 100 (74–156) n = 81 | 105 (76–126) n = 29 | 0.77 |

*(Continued)*

**Table 1.** (Continued)

| Variables | No FI | Mild FI | Moderate/Severe FI | p-value* |
|---|---|---|---|---|
| | (69.4%; n = 249) | (22.5%; n = 81) | (8.1%; n = 29) | |
| HDL (mg/dl) (Median; IQR 25% − 75%) n = 303 | 49.5 (40–57) n = 206 | 46 (39–61) n = 71 | 47 (43–57) n = 26 | 0.89 |
| LDL (mg/dl) (Median; IQR 25% − 75%) n = 302 | 111 (91–130) n = 206 | 114 (91–140) n = 71 | 105 (95–135) n = 25 | 0.78 |
| VLDL (mg/dl) (Median; IQR 25% − 75%) n = 350 | 20 (13–28) n = 242 | 20 (15–31) n = 79 | 21 (15–25) n = 29 | 0.84 |
| Glucose (mg/dl) (Median; IQR 25% − 75%) n = 358 | 97 (89–106) n = 248 | 96 (89–106) n = 81 | 93 (90–99) n = 29 | 0.40 |
| Quality of life-related domains (WHOQOL-BREF) | | | | |
| Physical health domain (Median; IQR 25% − 75%) | 64.3 (50–75) | 53.6 (39.3–64.3) | 50.0 (39.3–60.7) | <0.001 |
| Psychological domain (Median; IQR 25% − 75%) | 70.8 (58.3–79.2) | 66.7 (54.2–75) | 62.5 (50–66.7) | <0.001 |
| Social relationship domain (Median; IQR 25% − 75%) | 75.0 (58.3–83.3) | 66.7 (58.3–75) | 66.7 (58.3–75) | 0.001 |
| Environment domain (Median; IQR 25% − 75%) | 59.4 (53.1–68.7) | 50.0 (43.7–59.4) | 46.9 (40.6–53.1) | <0.001 |
| Overall domain (Median; IQR 25% − 75%) | 62.5 (62.5–75) | 62.5 (50–75) | 50.0 (37.5–62.5) | <0.001 |

† White (n=81), Black (n=49), Mulatto (n=216), Yellow (n=2), Indigenous (n=11).

‡ Other countries: patients from countries other than Brazil (Bolivia n = 1; Chile n = 1).

* Chi-squared test for categorical and Kruskal Wallis test for continuous variables.

** n=45 patients with both cardiac and digestive forms (cardiodigestive form).

FI = Food Insecurity; BMI = Body Mass Index; IQR = Interquartile Interval; HDL- high-density lipoprotein; LDL- low-density lipoprotein; VLDL- very-low-density lipoprotein.

## Discussion

Our study demonstrated that almost one-third of our urban cohort of CD chronic patients presented FI, most of them with mild FI (approximately 23%), but also with a considerable proportion of moderate/severe FI (around 8%). The presence of mild FI was associated with increased odds of obesity, while both mild and moderate/severe FI were associated with a decreased QoL across various domains, except for social relationship among those with moderate/severe FI. Collectively, these results call attention to the importance of FI in CD, a condition that affects a large number of individuals and has significant negative repercussions on their QoL.

The prevalence of FI in our study population was slightly lower than the 32.2% reported for individuals living in the Southeast region in the 2018 Family Budget Survey (POF) [31], a nationally representative survey conducted in Brazil during the same period. In this context, the lower frequency of FI in our study can be partially explained by the characteristics of our study population, which included patients regularly followed in a reference healthcare center that offers a comprehensive care including social assistant suport and appointments with nutritionist, which may have facilitated their connection to social support programs and better nutritional habits [32].

Furthermore, our results demonstrated an association between the presence of mild FI (but not moderate/severe FI) with a greater odds of obesity [33,34]. A possible explanation for this finding is that individuals with mild FI often rely on affordable food options, which are more likely to consist of ultra-processed and high-energy density products that have been linked to an elevated risk of obesity [14]. Prolonged FI can lead to metabolic adaptations in the body, such as reduced resting metabolic rate and increased fat storage, promoting weight gain particularly in situations of reduced physical activity influenced by low socioeconomic status [35–38].

In addition, FI is strongly linked to elevated levels of stress and anxiety that can increase cortisol release, a hormone that has been implicated in the development of obesity by increasing appetite and consumption of energy-dense,

**Table 2. Association between food insecurity status with comorbidities, anthropometric measures, biomarkers and QoL (n = 359).**

| Outcome Variables | Mild FI (*vs* No FI) | | Moderate/Severe FI (*vs* No FI) | |
|---|---|---|---|---|
| | (n = 81) | | (n = 29) | |
| | OR (95%CI) | | | |
| | Crude | Adjusted* | Crude | Adjusted* |
| Hypertension | 1.03 (0.61 to 1.77) | 1.16 (0.62 to 2.17) | 0.93 (0.41 to 2.09) | 0.68 (0.27 to 1.74) |
| Diabetes Mellitus | 1.00 (0.55 to 1.84) | 1.19 (0.63 to 2.26) | 0.73 (0.27 to 2.01) | 0.81 (0.28 to 2.34) |
| Dyslipidemia | 1.20 (0.72 to 1.99) | 1.34 (0.79 to 2.29) | 1.29 (0.59 to 2.83) | 1.50 (0.66 to 3.43) |
| **Obesity** | **1.82 (1.05 to 3.14)** | **1.83 (1.03 to 3.25)** | 1.31 (0.55 to 3.12) | 1.25 (0.50 to 3.13) |
| | β (95% CI) | | | |
| | Crude | Adjusted* | Crude | Adjusted* |
| BMI | +0.80 (−0.42 to +2.02) | +0.81 (−0.40 to +2.02) | −0.18 (−2.06 to +1.69) | −0.05 (−1.92 to +1.82) |
| Weight | +0.64 (−2.83 to +4.11) | +1.28 (−2.03 to +4.59) | −4.33 (−9.65 to +0.99) | −1.53 (−6.68 to +3.60) |
| Height | −0.01 (−0.35 to +0.007) | −0.006 (−0.02 to+0.01) | −0.04 (−0.07 to −0.007) | −0.01 (−0.04 to +0.01) |
| Waist Circumference | +0.56 (−2.43 to +3.55) | +1.15 (−1.77 to +4.09) | −2.59 (−7.17 to +1.99) | −1.24 (−5.79 to +3.31) |
| Total cholesterol | +2.32 (−6.84 to +11.48) | +2.15 (−7.10 to +11.41) | −3.48 (−17.51 to+10.55) | −4.31 (−18.67 to+10.05) |
| Triglycerides (n = 352) | +1.61 (−14.29 to+17.51) | +3.61 (−12.29 to +19.31) | −4.92 (−29.26 to +19.43) | −1.76 (−26.40 to+22.89) |
| HDL (n = 303) | +0.87 (−3.09 to +4.82) | +0.86 (−2.88 to +4.61) | +0.40 (−5.58 to +6.38) | −2.25 (−8.00 to +3.49) |
| LDL (n = 302) | +2.15 (−6.71 to +11.01) | +2.53 (−6.54 to +11.62) | +2.27 (−11.37 to +15.90) | +3.86 (−10.25 to +17.98) |
| VLDL (n = 350) | +0.26 (−2.69 to +3.21) | +0.68 (−2.26 to +3.63) | −0.59 (−5.07 to +3.87) | −0.20 (−4.72 to +4.32) |
| Glucose (n = 358) | +1.60 (−5.45 to +8.66) | +2.47 (−4.62 to +9.57) | −4.45 (−6.38 to +15.27) | +3.40 (−7.61 to +14.23) |
| Quality of life-related domains (WHOQOL-BREF) | β (95% CI) | | | |
| | Crude | Adjusted** | Crude | Adjusted** |
| **Physical health domain** | **−9.53 (−13.68 to −5.38)** | **−8.43 (−12.36 to −4.49)** | **−13.32 (−19.69 to −6.95)** | **−8.02 (−14.11 to −1.94)** |
| **Psychological domain** | **−6.43 (−10.19 to −2.66)** | **−5.54 (−9.11 to −1.96)** | **−11.69 (−17.46 to −5.92)** | **−7.22 (−12.76 to −1.68)** |
| **Social relationship domain** | **−6.62 (−10.30 to −2.95)** | **−6.62 (−10.37 to −2.88)** | −3.66 (−9.30 to +1.98) | −2.72 (−8.52 to +3.07) |
| **Environment domain** | **−9.37 (−12.53 to −6.21)** | **−8.79 (−11.99 to −5.60)** | **−15.74 (−20.59 to −10.89)** | **−13.56 (−18.51 to-8.62)** |
| **Overall domain** | **−9.05 (−13.03 to −5.08)** | **−8.10 (−12.15 to −4.06)** | **−18.77 (−24.87 to −12.68)** | **−16.82 (−23.08 to-10.57)** |

FI = Food Insecurity.

* Model adjusted by age, sex, race, schooling, cardiac and digestive form.

** Model adjusted by age, sex, race, schooling, hypertension, diabetes mellitus, dyslipidemia, obesity, cardiac and digestive form.

Estimates in bold are statistically significant (p < 0.05).

nutritionally poor foods [15,39,40]. In the presence of insulin, cortisol promotes triglyceride accumulation and retention in visceral fat depots which results in increased fat accumulation [41]. The existence of community and environmental factors, specifically food deserts, adds to the complexity of factors that facilitates weight gain [42]. On the other hand, individuals experiencing moderate/severe FI probably did not have minimal financial resources to afford even ultra-processed foods, which may have contributed to the absence of an association with obesity.

More recently, a study examining the association between FI and weight status among adults from a nationally representative sample of Brazil found that FI was associated with higher odds of obesity in women but lower odds in men. The observed sex differences in the relationship between FI and weight status may be driven by biological, behavioral, and sociocultural factors, which warrant further investigation to be fully elucidated [43]. The sample size of our study limited a more in-depth exploration of the relationship between FI and weight status by sex.

Consistent with previous findings, our study demonstrated an important association between all stages of FI (mild and moderate/severe FI) with lower QoL scores across several domains [16–18,39]. The association of FI with environmental

QoL domains can be explained by low financial resources and living conditions in deprived communities, that usually have precarious infrastructure [44]. These areas lack accessible food-based commerce and are devoid of large supermarkets, commonly referred as food deserts, which result in limited availability of affordable and nutritious food options [42].

In terms of the physical QoL domain, FI significantly affects the availability of nutrients, leading to increased malnutrition and potentially resulting in decreased stamina [45,46]. Moreover, living in a food desert compounds the challenges faced in physically accessing major stores. The combination of decreased stamina, the constant effort required to obtain food, and limited accessibility to essential stores contributes to the association of FI with the physical QoL domain [42,44].

Furthermore, previous studies indicate that individuals experiencing FI often encounter elevated levels of stress, anxiety, and feeling of shame [15,47,48]. These factors can significantly influence their perception of psychological well-being and social relationships. The feeling of shame, in particular, could be a factor in decreasing social relationships in those experiencing FI. Consequently, all these factors together can lead to a decline in overall QoL [16,17,39].

Conversely, an intriguing observation of our study was the lack of association between moderate/severe FI and social relationship QoL domain. One possible explanation is that individuals facing moderate/severe FI may receive social support from their families and communities, characterized by high levels of social and family cohesion, and social support, decreasing the association of FI with their QoL [49–51].

The present study had some limitations that should be acknowledged. First, the cross-sectional design restricts the establishment of causality or determination of the temporal relationship between FI, comorbidities, and QoL. Furthermore, the provision of high-quality healthcare services by our institution, recognized as a national reference center for infectious disease treatment, may have mitigated the association between FI and comorbidities. Although the study accounted for various sociodemographic and clinical covariates, there may still be unmeasured confounding factors, such as psychological and social aspects (e.g., participation in social support programs), which were not explicitly considered and could impact the observed associations. Moreover, despite we have used the validated EBIA scale available at the time of study initiation in 2014 [21], a more recent version was validated afterward [52], offering an updated instrument for assessing FI in a Brazilian population. Finally, we assessed FI at the individual rather than the household level. However, it is well established that within food-insecure households, adults typically experience FI first, as food allocation often prioritizes children. Therefore, although FI is mostly measured at the household level or at individual level considering the head of the family, its consequences are directly experienced by all individual household members [53,54]. This supports the evaluation of associations between FI and individual characteristics, such as anthropometric measures, lipid profile and blood glucose, comorbidities, and QoL. On the other hand, the strength of our cohort lies in its representation of patients from diverse regions across Brazil, making it representative of the population with CD in the country [20].

To conclude, the prevalence of FI among individuals with CD was similar to the rates reported in Brazilian representative national surveys, with mild FI being more common. Both mild and moderate/severe FI were associated with lower QoL across several domains, with significant negative association on physical health, psychological well-being, social relationships, and the overall QoL. Intervention strategies aiming to improve FI status and its impact on various aspects of well-being are needed.

## Supporting information

**S1 Table. Clinical and demographic characteristics of participants (n = 359).**
(DOCX)

## Author contributions

**Conceptualization:** Paula Simplício da Silva, Roberto Magalhães Saraiva, Alejandro Marcel Hasslocher-Moreno, Mauro Felippe Felix Mediano.

**Data curation:** Mauro Felippe Felix Mediano.

**Formal analysis:** Celson Júnio do Nascimento Costa, Tatiana Rehder Gonçalves, Luiz Fernando Rodrigues Junior, Mauro Felippe Felix Mediano.

**Funding acquisition:** Roberto Magalhães Saraiva, Mauro Felippe Felix Mediano.

**Investigation:** Celson Júnio do Nascimento Costa, Roberto Magalhães Saraiva, Luiz Henrique Conde Sangenis, Marcelo Teixeira de Holanda, Gilberto Marcelo Sperandio da Silva, Isis Gabrielli Gomes Xavier, Henrique Silveira Costa, Tatiana Rehder Gonçalves, Grazielle Vilas Boas Huguenin, Alejandro Marcel Hasslocher-Moreno, Daniel Arthur Barata Kasal, Mauro Felippe Felix Mediano.

**Methodology:** Roberto Magalhães Saraiva, Fernanda de Souza Nogueira Sardinha Mendes, Isis Gabrielli Gomes Xavier, Daniel Arthur Barata Kasal, Mauro Felippe Felix Mediano.

**Project administration:** Mauro Felippe Felix Mediano.

**Resources:** Mauro Felippe Felix Mediano.

**Software:** Mauro Felippe Felix Mediano.

**Supervision:** Mauro Felippe Felix Mediano.

**Validation:** Mauro Felippe Felix Mediano.

**Visualization:** Mauro Felippe Felix Mediano.

**Writing – original draft:** Celson Júnio do Nascimento Costa, Paula Simplício da Silva, Roberto Magalhães Saraiva, Luiz Henrique Conde Sangenis, Marcelo Teixeira de Holanda, Gilberto Marcelo Sperandio da Silva, Fernanda de Souza Nogueira Sardinha Mendes, Isis Gabrielli Gomes Xavier, Henrique Silveira Costa, Tatiana Rehder Gonçalves, Luiz Fernando Rodrigues Junior, Cristiane da Cruz Lamas, Grazielle Vilas Boas Huguenin, Alejandro Marcel Hasslocher-Moreno, Daniel Arthur Barata Kasal, Mauro Felippe Felix Mediano.

**Writing – review & editing:** Celson Júnio do Nascimento Costa, Paula Simplício da Silva, Roberto Magalhães Saraiva, Luiz Henrique Conde Sangenis, Marcelo Teixeira de Holanda, Gilberto Marcelo Sperandio da Silva, Fernanda de Souza Nogueira Sardinha Mendes, Isis Gabrielli Gomes Xavier, Henrique Silveira Costa, Tatiana Rehder Gonçalves, Luiz Fernando Rodrigues Junior, Cristiane da Cruz Lamas, Grazielle Vilas Boas Huguenin, Alejandro Marcel Hasslocher-Moreno, Daniel Arthur Barata Kasal, Mauro Felippe Felix Mediano.

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
