## [Decision Letter · Decision Letter 0]

PONE-D-24-09213Food insecurity is associated with decreased quality of life in patients with chronic Chagas diseasePLOS ONE

Dear Dr. Mediano,

Thank you for submitting your manuscript to PLOS ONE. After careful consideration, we feel that it has merit but does not fully meet PLOS ONE’s publication criteria as it currently stands. Therefore, we invite you to submit a revised version of the manuscript that addresses the points raised during the review process.

We look forward to receiving your revised manuscript.

Kind regards,

Karina Cardoso Meira, Ph.D

Academic Editor

PLOS ONE

“None to declare.”

4. In the online submission form, you indicated that [The data underlying the results presented in the study are available upon reasonable request to the corresponding author.].

Additional Editor Comments:

The study makes a significant contribution to the discourse on social inequalities among individuals affected by Chagas disease (CD). It articulates a relevant objective and adheres to the necessary ethical requirements. However, several limitations warrant clarification, particularly regarding methodological issues and the discussion of the study, as highlighted by the reviewers.

Reviewers' comments:

Reviewer's Responses to Questions

**Comments to the Author**

1. Is the manuscript technically sound, and do the data support the conclusions?

Reviewer #1: No

Reviewer #2: Yes

2. Has the statistical analysis been performed appropriately and rigorously? 

Reviewer #1: No

Reviewer #2: Yes

3. Have the authors made all data underlying the findings in their manuscript fully available?

Reviewer #1: Yes

Reviewer #2: Yes

4. Is the manuscript presented in an intelligible fashion and written in standard English?

Reviewer #1: Yes

Reviewer #2: Yes

5. Review Comments to the Author

Reviewer #1: The study makes an important contribution to the debate on social inequalities in individuals with Chagas disease (CD). It also presents a relevant objective and the necessary ethical requirements. However, there are some points that deserve to be highlighted:

- The introduction does not explore the health inequities of patients with CD in Brazil and Latin America. It focuses on health indicators in a broad way, not exposing specificities of the reality and context of the study population.

- As it was a study with a cross-sectional epidemiological design, the data collection time was very long.

- What is the maximum age limit? Elderly people too? From the perspective of FI and quality of life, the elderly would have different debates.

- The magnitude of the INI (Evandro Chagas National Institute of Infectious Diseases) for patients with CD in Brazil is not clear in the methodology. What does this mean, thinking about readers who do not know the reality of INI? Does it allow inferences? Local level? National?

- The authors need to better explain the inclusion, exclusion and sampling criteria, if applicable. How do patients get to the INI?

- Why two months between visits? Was there a minimum period between visits?

- Regarding food insecurity (FI), the article presents important methodological issues:

(a) The recommendation in Brazil for the use of EBIA is the questionnaire with a scale of 14 questions, validated in 2014 (https://www.scielo.br/j/rn/a/X9vkr9sc7WX8tH8dcWP8XPN/?format=pdf&lang=en) and not the one used in the study. In some contexts, the short 8-question questionnaire may be used. All commonly used in IBGE research, for example.

(b) The dimension of the EBIA for measuring FI levels is household and not individual. Individual variables can be related to FI, if it belongs to the head of the household. And there is no mention of this throughout the text.

(c) Considering the FI outcome, I find the associations made by the authors to be delicate, such as total cholesterol (mg/dl) and weight or height in isolation. What is the real objective of relating a possible experience of hunger at home with an individual's cholesterol level? What about the weight or height ratio of adults, without a relationship like BMI? Why would it be associated with FI? These are examples of just some of the variables. But there are others where the association would not make sense.

- Another important point in the methodology concerns statistical analyses: the authors carried out logistic regression models to estimate OR. What binary outcome? FI Yes or no? Because in table 2, there are 3 outcomes (FI, which I understand to be the reference, Mild FI and Moderate/Severe FI). Therefore, you would need a multinomial model.

- Due to methodological limitations, all results are dubious inferences and should be redone. The same applies to the discussion.

Reviewer #2: Parabenizo os autores por abordarem as principais lacunas na temática das pessoas com Doença de Chagas (DC) e por ressaltarem a importância das pesquisas nesse campo. No entanto, recomendo a reformulação de alguns parágrafos para aprimorar a coesão e o refinamento do texto.

Linha 64-65: Sugiro focar na América Latina e, se possível, abordar os desafios do atendimento a pessoas com DC no Brasil.

Linha 65-69: Recomendo a reformulação da frase para melhorar a abordagem da relação entre DC, condições de vida e acesso à alimentação. Indago qual é a relação entre o aumento da expectativa de vida e a migração; alguns estudos apontam que, apesar da baixa renda e escolaridade, existe uma longevidade atrelada às condições de vida na área rural. O aumento da "exposição a estilos de vida pouco saudáveis" seria prejudicial para os pacientes ou para os moradores da área urbana?

Linha 72-75: Na frase “The factors that contribute to the link between CD and poverty are complex, including poor housing conditions and inadequate access to healthcare services, that are more commonly observed in middle and low-income countries.”, sugiro a adição de uma referência.

Linha 78-79: Recomendo inserir uma conexão entre os parágrafos.

Linha 79-90: Sugiro dividir o parágrafo, criando um dedicado à Insegurança Alimentar (IA) e às doenças possivelmente associadas. Sugiro criar outro parágrafo a partir da frase “Considering the low socioeconomic status of individuals with CD,...”

Linha 123-133: As escalas de aferição da IA são distintas entre países, e no Brasil foi validada a EBIA. Desde 2004 foram realizadas algumas alterações na escala, e atualmente a versão indicada para o uso é a de 14 itens (Segall-Corrêa AM, Marin-León L, Melgar-Quiñonez H, Pérez-Escamilla R. Refinement of the Brazilian Household Food Insecurity Measurement Scale: Recommendation for a 14-item EBIA. Rev Nutr. abril de 2014;27:241–51.). Apesar da comparabilidade entre as escalas, autores realizam adaptação das escalas (como nos resultados dos relatório da Pesquisa Nacional de Insegurança Alimentar durante a Pandemia de Covid-19 realizada pela Rede Penssan). Caso não seja possível a reformulação nesta pesquisa, indico que nas próximas pesquisas, seja utilizada a versão mais atual da EBIA completa (14 itens).

Linha 167-186: Recomendo descrever separadamente as variáveis que caracterizam tanto o domicílio quanto o paciente. A raça/cor da pessoa é um dos principais indicadores da IA. No entanto, sugiro evitar a associação de grupos como pretos, pardos, indígenas e amarelos, pois isso pode levar à superestimação dos dados, considerando as diferentes prevalências de IA em cada grupo populacional. Proponho a união de pretos e pardos em uma categoria denominada "negros" (em português; em inglês, utiliza-se "Black/Brown"). Questiono também quantos pacientes indígenas e amarelos foram incluídos no estudo, uma vez que outros trabalhos coletaram esses dados, mas não os utilizaram nas análises devido ao baixo número amostral e ao risco de viés.

Resultados: Sugiro adicionar uma tabela, em anexo, das características dos pacientes sem a estratificação pela IA. Avaliar a possibilidade com os demais autores.

Table 1: Indico inserir o intervalo de confiança. Sugiro que os valores fiquem dispostos desta forma: % (n; IC95%). Avaliar a necessidade de apresentar o número da amostra. Indico elucidar o que seriam “Other countries”, visto que não está disposto na legenda e na metodologia – ainda indico rever a necessidade da sua permanecência na tabela.

Linha 252 – 261: Trago uma reflexão para os autores. A EBIA é uma escala de percepção da IA, não avalia quantitativamente – apesar da utilização das prevalências. Os dados da IA do ano de 2013 (PNAD) e 2018 (POF) são diferentes devido o cenário de crises políticas, econômicas, ambientais e sociais, e pergunto-me como os dados obtidos no estudo foram semelhante as pesquisas, se estas mesmas possuem dados distintos. Além disso, embora os programas de transferência de renda sejam extremamente importantes, os valores estão defasados em relação ao cenário de desigualdades sociais e crises sistêmicas no país, podendo não garantir a segurança alimentar nos domicílios. Com isso, sugiro relacionar a prevalência da IA aos indicadores sociais identificados na amostra, que podem influenciar o acesso a uma alimentação adequada e saudável (como escolaridade, rendimento, macrorregião, dentre outros).

Linha 262-277: Indico diminuir ou dividir o parágrafo – está extenso. Existem poucos estudos no Brasil que correlacionem a IA e estado nutricional, por isso, parabenizo pela discussão. Indico este artigo, por ser um dos mais recentes nacionalmente na temática: Domingos, T. B., Sichieri, R., & Salles-Costa, R. (2022). Sex differences in the relationship between food insecurity and weight status in Brazil. The British journal of nutrition, 1–19. Advance online publication. https://doi.org/10.1017/S0007114522001192

Linha 278-299: Indico diminuir ou dividir o parágrafo – está extenso.

6. PLOS authors have the option to publish the peer review history of their article (what does this mean? ). If published, this will include your full peer review and any attached files.

**Do you want your identity to be public for this peer review?** For information about this choice, including consent withdrawal, please see our Privacy Policy .

Reviewer #1: No

Reviewer #2: No

---

## [Author Response · Author response to Decision Letter 1]

26 Mar 2025

Reviewer #1:

1- The introduction does not explore the health inequities of patients with CD in Brazil and Latin America. It focuses on health indicators in a broad way, not exposing specificities of the reality and context of the study population.

Response: We thank the reviewer for this suggestion. We have included more detailed information in the introduction section, better exploring the specificities and the current health challenges faced by patients with CD.

2- As it was a study with a cross-sectional epidemiological design, the data collection time was very long.

Response: Despite we agree with the reviewer that data collection period was long, no major changes in the epidemiological profile have occurred during this period, therefore does not impacting the quality of our results, neither the evaluated associations. Even in a reference center, recruiting patients to participate in a study is sometimes challenging, which can take more time than initially assumed.

3- What is the maximum age limit? Elderly people too? From the perspective of FI and quality of life, the elderly would have different debates.

Response: Chagas disease population is aging over the last decades and the inclusion of older individuals better reflects the study population. To account for potential effects of age on the outcomes, we included age as a potential confounder in our statistical models, ensuring that our findings were adjusted for this factor.

4- The magnitude of the INI (Evandro Chagas National Institute of Infectious Diseases) for patients with CD in Brazil is not clear in the methodology. What does this mean, thinking about readers who do not know the reality of INI? Does it allow inferences? Local level? National?

Response: INI-Fiocruz is a national reference center for the treatment and research of infectious and tropical diseases within the Brazilian National Health Service (Sistema Único de Saúde - SUS), operating under the Brazilian Ministry of Health. INI-Fiocruz receives patients from various regions across the country, providing comprehensive, multidisciplinary care to patients with CD. This information was included in the manuscript.

5 - The authors need to better explain the inclusion, exclusion and sampling criteria, if applicable. How do patients get to the INI?

Response: We rephrased this sentence in the manuscript to improve clarity.

6- Why two months between visits? Was there a minimum period between visits?

Response: Two months was the period that we determined as reasonable to accurately capture information without major changes in the variables (exposure, outcome, and covariates) of the study. There was no minimum period between visits.

7- Regarding food insecurity (FI), the article presents important methodological issues:

(a) The recommendation in Brazil for the use of EBIA is the questionnaire with a scale of 14 questions, validated in 2014 (https://www.scielo.br/j/rn/a/X9vkr9sc7WX8tH8dcWP8XPN/?format=pdf&lang=en) and not the one used in the study. In some contexts, the short 8-question questionnaire may be used. All commonly used in IBGE research, for example.

Response: We appreciate the reviewer for this important comment. At the time our study was designed (2013), the 14-question version of the Brazilian Food Insecurity Scale (EBIA), which was validated in 2014, had not yet been published. Therefore, we used the previous version, which was available and established for research use at that time, as referenced in the methods section (https://doi.org/10.1093/jn/134.8.1923). We have included this information in the limitation section of the manuscript.

(b) The dimension of the EBIA for measuring FI levels is household and not individual. Individual variables can be related to FI, if it belongs to the head of the household. And there is no mention of this throughout the text.

Response: We acknowledge that EBIA assesses FI mostly at the household level or at individual level considering the head of the family. However, it is well established that within food-insecure households, adults typically experience FI first, as food allocation often prioritizes children. Therefore, while EBIA measures FI at the domicile level, its consequences are directly experienced by all individual household members. This supports the evaluation of associations between FI and individual characteristics, such as anthropometric measures, lipid profile and blood glucose, comorbidities, and QoL. We have clarified this point in the manuscript and will ensure that the relationship between household FI and individual-level outcomes is explicitly addressed.

(c) Considering the FI outcome, I find the associations made by the authors to be delicate, such as total cholesterol (mg/dl) and weight or height in isolation. What is the real objective of relating a possible experience of hunger at home with an individual's cholesterol level? What about the weight or height ratio of adults, without a relationship like BMI? Why would it be associated with FI? These are examples of just some of the variables. But there are others where the association would not make sense.

Response: The associations between FI and anthropometric, clinical, and QoL variables have been explored in previous studies, as noted in the introduction (3rd paragraph). The rationale for examining these associations relied on the relationship between FI and poorer dietary quality, often characterized by a higher intake of calorie-dense, nutrient-poor foods. This dietary pattern can contribute to unfavorable anthropometric and lipid profiles, increased comorbidities, and lower QoL. Regarding the association between FI and height, while our study is cross-sectional, a potential relationship between FI and shorter stature may indicate long-term nutritional deficiencies, particularly when FI occurs during critical growth periods. The investigation of the relationship between FI and anthropometric, clinical, and QoL is especially relevant in a population of patients with CD, as their lower socioeconomic status may make them more vulnerable to FI and its associated health consequences.

8- Another important point in the methodology concerns statistical analyses: the authors carried out logistic regression models to estimate OR. What binary outcome? FI Yes or no? Because in table 2, there are 3 outcomes (FI, which I understand to be the reference, Mild FI and Moderate/Severe FI). Therefore, you would need a multinomial model.

Response: In our study, FI was the exposure, while anthropometric measurements, lipid profile, blood glucose, comorbidities, and QoL were the outcomes. As described in the manuscript, we used logistic regression models to assess the association between FI categories (exposure) and the presence of comorbidities (arterial hypertension, diabetes mellitus, dyslipidemia, and obesity), which is appropriate since these are binary outcomes. For continuous outcomes (anthropometric measurements, lipid profile, blood glucose, and QoL), we applied linear regression models. To improve clarity, we have rephrased the relevant section in the statistical methods.

9 - Due to methodological limitations, all results are dubious inferences and should be redone. The same applies to the discussion.

Response: We appreciate the reviewer for this comment. We made all the major changes required by the reviewer. The statistical models that we used in our data analysis are correct, and we rephrased the relevant section in the statistical methods to improve text comprehension.

Reviewer 2

1- Linha 64-65: Sugiro focar na América Latina e, se possível, abordar os desafios do atendimento a pessoas com DC no Brasil.

Response: We made the change accordingly.

2- Linha 65-69: Recomendo a reformulação da frase para melhorar a abordagem da relação entre DC, condições de vida e acesso à alimentação. Indago qual é a relação entre o aumento da expectativa de vida e a migração; alguns estudos apontam que, apesar da baixa renda e escolaridade, existe uma longevidade atrelada às condições de vida na área rural. O aumento da "exposição a estilos de vida pouco saudáveis" seria prejudicial para os pacientes ou para os moradores da área urbana?

Response: We appreciate the reviewer for this suggestion. We rephrased the sentence to improve clarity as follows: “Additionally, the migration of a significant proportion of the CD population from rural to urban areas has led to greater exposure to unhealthy lifestyles, which may contribute to the development of several non-communicable diseases, such as obesity, hypertension, diabetes mellitus, and dyslipidemia, while negatively affecting their quality of life (QoL)”.

3 - Linha 72-75: Na frase “The factors that contribute to the link between CD and poverty are complex, including poor housing conditions and inadequate access to healthcare services, that are more commonly observed in middle and low-income countries.”, sugiro a adição de uma referência.

Response: A reference has been added to this sentence.

4- Linha 78-79: Recomendo inserir uma conexão entre os parágrafos.

Response: We have included a sentence to better connect these paragraphs, as suggested.

5- Linha 79-90: Sugiro dividir o parágrafo, criando um dedicado à Insegurança Alimentar (IA) e às doenças possivelmente associadas. Sugiro criar outro parágrafo a partir da frase “Considering the low socioeconomic status of individuals with CD,...”.

Response: We rephrased the entire paragraph to improve clarity.

6- Methods (FI): Linha 123-133: As escalas de aferição da IA são distintas entre países, e no Brasil foi validada a EBIA. Desde 2004 foram realizadas algumas alterações na escala, e atualmente a versão indicada para o uso é a de 14 itens (Segall-Corrêa AM, Marin-León L, Melgar-Quiñonez H, Pérez-Escamilla R. Refinement of the Brazilian Household Food Insecurity Measurement Scale: Recommendation for a 14-item EBIA. Rev Nutr. abril de 2014;27:241–51.). Apesar da comparabilidade entre as escalas, autores realizam adaptação das escalas (como nos resultados dos relatório da Pesquisa Nacional de Insegurança Alimentar durante a Pandemia de Covid-19 realizada pela Rede Penssan). Caso não seja possível a reformulação nesta pesquisa, indico que nas próximas pesquisas, seja utilizada a versão mais atual da EBIA completa (14 itens).

Response: We appreciate the reviewer for this important comment. At the time our study was designed (2013), the 14-question version of the Brazilian Food Insecurity Scale (EBIA), which was validated in 2014, had not yet been published. Therefore, we used the previous version, which was available and established for research use at that time, as referenced in the methods section (https://doi.org/10.1093/jn/134.8.1923). We have included this information in the limitation section of the manuscript.

7 - Methods (Covariates): Linha 167-186: Recomendo descrever separadamente as variáveis que caracterizam tanto o domicílio quanto o paciente. A raça/cor da pessoa é um dos principais indicadores da IA. No entanto, sugiro evitar a associação de grupos como pretos, pardos, indígenas e amarelos, pois isso pode levar à superestimação dos dados, considerando as diferentes prevalências de IA em cada grupo populacional. Proponho a união de pretos e pardos em uma categoria denominada "negros" (em português; em inglês, utiliza-se "Black/Brown"). Questiono também quantos pacientes indígenas e amarelos foram incluídos no estudo, uma vez que outros trabalhos coletaram esses dados, mas não os utilizaram nas análises devido ao baixo número amostral e ao risco de viés.

Response: We appreciate the reviewer’s comments. We have re-categorized race as White and Non-White (grouping Black, Mulatto together with Yellow, and Indigenous) due to the small number of participants who self-identified as Yellow (n=2) or Indigenous (n=11), making it difficult to consider Yellow and Indigenous as a separate group. However, given their representativeness in our study population, we chose not to exclude them. We conducted a sensitivity analysis excluding Yellow and Indigenous participants and the results remained unchanged. We have included the number of participants for each race (white, black, mulatto, yellow and Indigenous) in the footnote of Table 1.

8 - Resultados: Sugiro adicionar uma tabela, em anexo, das características dos pacientes sem a estratificação pela IA. Avaliar a possibilidade com os demais autores.

Table 1: Indico inserir o intervalo de confiança. Sugiro que os valores fiquem dispostos desta forma: % (n; IC95%). Avaliar a necessidade de apresentar o número da amostra. Indico elucidar o que seriam “Other countries”, visto que não está disposto na legenda e na metodologia – ainda indico rever a necessidade da sua permanecência na tabela.

Response: We have included Supplemental Table 1, which presents the overall characteristics of the study participants. The variable "Other countries" refers to patients from countries other than Brazil (Bolivia: n=1; Chile: n=1), and this information has been added to the footnote of Table 1.

Regarding the suggestion to include 95% CIs in Table 1, our primary aim with Table 1 is to summarize the descriptive characteristics of the study population rather than to infer associations or estimate population parameters. Therefore, we have chosen to present the data using standard descriptive statistics (e.g., median and interquartile range for continuous variables and counts/percentages for categorical variables).

Additionally, since inferential analyses—including 95% CIs—are already provided in the regression models (Table 2), we believe that adding CIs to Table 1 would not provide additional meaningful information to the manuscript.

9- Discussion: Linha 252 – 261: Trago uma reflexão para os autores. A EBIA é uma escala de percepção da IA, não avalia quantitativamente – apesar da utilização das prevalências. Os dados da IA do ano de 2013 (PNAD) e 2018 (POF) são diferentes devido o cenário de crises políticas, econômicas, ambientais e sociais, e pergunto-me como os dados obtidos no estudo foram semelhante as pesquisas, se estas mesmas possuem dados distintos. Além disso, embora os programas de transferência de renda sejam extremamente importantes, os valores estão defasados em relação ao cenário de desigualdades sociais e crises sistêmicas no país, podendo não garantir a segurança alimentar nos domicílios. Com isso, sugiro relacionar a prevalência da IA aos indicadores sociais identificados na amostra, que podem influenciar o acesso a uma alimentação adequada e saudável (como escolaridade, rendimento, macrorregião, dentre outros).

Response: We appreciate the reviewer for this insightful comment. We have rephrased the sentence to compare our results specifically with the FI percentage from the Southeast region in the 2018 POF, as this population shares more similar characteristics with our study sample than the national average.

10 - Linha 262-277: Indico diminuir ou dividir o parágrafo – está extenso. Existem poucos estudos no Brasil que correlacionem a IA e estado nutricional, por isso, parabenizo pela discussão. Indico este artigo, por ser um dos mais recentes nacionalmente na temática: Domingos, T. B., Sichieri, R., & Salles-Costa, R. (2022). Sex differences in the relationship between food insecurity and weight status in Brazil. The British journal of nutrition, 1–19. Advance online publication. https://doi.org/10.1017/S0007114522001192

Response: We appreciate the reviewer's suggestion. We have divided the paragraph to enhance readability and have incorporated the suggested reference in the discussion section.

11 - Linha 278-299: Indico diminuir ou dividir o parágrafo – está extenso.

Response: Thanks for your recommendation. We made the change accordingly.

---

## [Decision Letter · Decision Letter 1]

Food insecurity is associated with decreased quality of life in patients with chronic Chagas disease

PONE-D-24-09213R1

Dear Dr.Mediano,

We’re pleased to inform you that your manuscript has been judged scientifically suitable for publication and will be formally accepted for publication once it meets all outstanding technical requirements.

Kind regards,

Karina Cardoso Meira, Ph.D

Academic Editor

PLOS ONE

Additional Editor Comments (optional):

Dear Dr Mediano.

Congratulations on the thoughtful revisions you have made to the manuscript. Both reviewers appreciate the quality of your work and its relevance to understanding the clinical and demographic correlates of food insecurity.

Reviewers' comments:

Reviewer's Responses to Questions

**Comments to the Author**

1. If the authors have adequately addressed your comments raised in a previous round of review and you feel that this manuscript is now acceptable for publication, you may indicate that here to bypass the “Comments to the Author” section, enter your conflict of interest statement in the “Confidential to Editor” section, and submit your "Accept" recommendation.

Reviewer #1: All comments have been addressed

Reviewer #2: All comments have been addressed

2. Is the manuscript technically sound, and do the data support the conclusions?

Reviewer #1: Yes

Reviewer #2: Yes

3. Has the statistical analysis been performed appropriately and rigorously? 

Reviewer #1: Yes

Reviewer #2: Yes

4. Have the authors made all data underlying the findings in their manuscript fully available?

Reviewer #1: Yes

Reviewer #2: Yes

5. Is the manuscript presented in an intelligible fashion and written in standard English?

Reviewer #1: Yes

Reviewer #2: Yes

6. Review Comments to the Author

Reviewer #1: I find the responses satisfactory and I think the authors were careful in reviewing the requests made.

Reviewer #2: I congratulate the authors for the changes in the manuscript.

Results:

# Table 1 contains the clinical and demographic characteristics according to Food Insecurity. The results are interesting. However, when analyzing other articles that work with the theme, the values are distributed in the row (and not in the column). This occurs because the covariates are usually analyzed based on the levels of food insecurity or food security.

In fact, the authors point out the distribution in the row in the study population: Food Security = 69.4%; n=249; Mild Food Insecurity = 22.5%; n=81) and Moderate/Severe Food Insecurity = 8.1%; n=29.

I recommend caution when comparing in the discussion with other articles, since the prevalence may have been analyzed in another format.

Possibly, due to the analysis in the column, some results differ from the general population:

- 7% of people with more education in Moderate/Severe Food Insecurity;

- The Northeast has the highest Food Security when compared to the other regions (67.1%);

- 48.3% of eutrophic people in Moderate/Severe Food Insecurity and 45.5% in Food Security.

If possible, I suggest that an analysis be carried out to confirm whether the best approach would not be the analysis in the row, considering the levels of insecurity, rather than the distribution in the column (considering the categories).

# If possible, I suggest that you divide Table 1 into two. One for prevalence and another for averages or divide Table 1 for demographic characteristics and Table 2 for clinical characteristics.

Besides, I have nothing else to add. I hope that the comments will enrich the manuscript.

7. PLOS authors have the option to publish the peer review history of their article (what does this mean? ). If published, this will include your full peer review and any attached files.

**Do you want your identity to be public for this peer review?** For information about this choice, including consent withdrawal, please see our Privacy Policy .

Reviewer #1: No

Reviewer #2: No

---

## [Editor Report · Acceptance letter]

PONE-D-24-09213R1

PLOS ONE

Dear Dr. Mediano,

I'm pleased to inform you that your manuscript has been deemed suitable for publication in PLOS ONE. Congratulations! Your manuscript is now being handed over to our production team.

Kind regards,

on behalf of

Dr. Karina Cardoso Meira

Academic Editor

PLOS ONE